materials science/nanotechnology

molecular dynamics, ReaxFF force field, adsorption

**Author for correspondence:**
Junpeng Liu
e-mail: pureindigo@hrbeu.edu.cn

# Adsorption of ethanol molecules on the Al (1 1 1) surface: a molecular dynamic study

## Pingan Liu, Junpeng Liu and Mengjun Wang

College of Aerospace and Civil Engineering, Harbin Engineering University, Harbin City, Heilongjiang Province 150001, People's Republic of China

(iD) JL, 0000-0001-9649-7143

The adsorption process of ethanol molecules on Al slabs was investigated by molecular dynamic simulations with a ReaxFF force field. The force field used in this paper has been validated by comparing adsorption energy results with quantum mechanical (QM) calculations. All simulations were performed under the canonical (NVT) ensemble. The single-molecule adsorption simulation shows that the hydroxyl group plays a more important role in the whole progress than the ethyl group. Besides, decomposition of hydroxyl groups was also observed during multimolecule adsorption processes. Simulations of adsorption processes of Al slab by ethanol molecules at different temperatures and pressures (controlled by the number of ethanol molecules) was also performed. System energy and radial distribution function (RDF) plots were invoked to describe adsorption processes and centro-symmetry parameter (CSP) analysis was adopted to study the surface properties with coating layers. Our results indicate that the whole adsorption process can be divided into two periods and the greater the pressure, the more ethanol molecules diffuse into the Al slab. How raising the temperature helps the adsorption processes is related to the initial number of molecules. The crystal structure of the Al surface will become amorphous under the constant impact of ethanol molecules.

## 1. Introduction

In recent years, nanometallic materials have attracted great research interest for their outstanding physical and chemical properties in comparison with macroscopic materials [1]. Among numerous applications of nano-sized metal materials to improve the properties of original materials, Al has been proved to be an effective additive for solid propellants. In fact, Al has been extensively used in many propulsion systems including traditional

solid propulsion pillars and underwater propellants. The addition of Al nanoparticles (ANP) brings propellants' higher energy output, burning rate and lower ignition temperature [2,3]. However, for every plus there is a minus, and the high chemical reactivity of ANPs also triggers a series of problems: ANPs are vulnerable when exposed to an oxygen environment. Related research studies show that compared with micron-sized Al powders, the weight content of aluminium oxide for most ANPs exceeds 10%, whereas that of micro-sized Al powders is only 1.5–5% [4–6]. Moreover, it is another challenge to properly store ANPs and maintain their chemical activity for a long period of time. Take an ANP with 3 nm thick oxide shell as an example: the mass fraction of the futile substance ($Al_2O_3$) increases with decreasing particle size dramatically; the value can reach as high as 52% when the particle size decreases to 38 nm. To overcome these defects, one method is to coat ANPs with a protection layer. By now, there are two methods of coating nascent ANPs. The first method is using inorganic substances which include the metal and their oxide and metal salt compound. The second method is coating ANPs with organic materials that have shown more advantages in various aspects. The coating layer with organic matter can provide extra heat when ANPs are ignited but maintain anti-oxidation properties at low temperature. Compared with their metallic counterparts, which are usually produced by high-energy laser irradiation, the organic components dissolved in corresponding solvent can coat ANPs perfectly after sufficient mechanical agitation which is quite energy saving. Besides, organic compounds with certain terminated groups show better hydrophobicity and compatibility with other ingredients than metal coatings [6–9].

Gromov *et al.* used various hydrocarbon precursors, including nitrocellulose, oleic acid, stearic acid and kerosene, to coat ANPs [4–6]. Their work showed that, with organic coatings, ANPs' enthalpy of combustion will be improved and the content of the metal in ANPs also increased, which means that ANPs were entirely coated and active metal content was preserved. Sossi *et al.* conducted similar experiments, but their results revealed that oxide layers were present under organic layers [10]. Huang *et al.* successfully coated ANPs with hydroxyl-terminated polybutadiene (HTPB) and dioctyl sebacate (DOS) both of which are typical substances for solid propellants [11,12]. Their works showed that the ANPs coated by such hydrophobic substances exhibit improved oxidation resistance.

In the above experiments, ethanol is usually used as the main component of solvent or protective solution for long-time storage. Although many experiments have been done, only a few studies interpret interactions between the Al substrate and coating material from a molecular perspective. Molecular dynamics (MD) simulation is a powerful tool to unravel the mechanisms of ethanol adsorption on Al and can provide detailed structural information. MD simulation has been successfully applied in molecule adsorption. Li *et al.* performed steered molecular dynamics simulations to study the effect of the degree of substrate ionization on adsorption stability [13]. He *et al.* studied interactions between various amino acids and carbon nanotubes and pointed out that the electrostatic attractions play an important role in the adsorption process [14]. Su *et al.* show that electrostatics and surface functional groups affect the adhesion strengths in an interdependent relationship [15]. The above studies prove that MD simulation is especially useful for studying the influence of charge and microstructure analysis.

In this paper, we performed MD simulations in the canonical (NVT) ensemble to simulate the adsorption of ethanol molecules on the Al slab by using a reactive force field. Our research focuses on the isotherm adsorption process and the key role of hydroxyl groups in the adsorption process. The rest of this paper is arranged as follows: the third chapter introduces the force field and simulation settings. Force field verification process, adsorption process and surface properties are discussed in the fourth chapter.

# 2. Material and methods

## 2.1. QM method

To verify whether the ReaxFF force field, taken from [7], can provide a reasonable description of the interaction between Al and ethanol molecules, we carried out QM calculations to test the force field. According to [7], the force field has been proven to have the ability to describe quantitatively reaction kinetics between hydrocarbon radicals and Al atoms. Therefore, our major concern in this paper is to test the interactions between the hydroxyl substance and aluminium atoms. The settings of QM calculations are as follows. The gradient-corrected functional (GGA-PBE) was used to represent the exchange-correlation potential used in the calculation, and a maximum cut-off value (400 eV) was chosen for the plane wave basis set. Three layers of Al (1 1 1) slab (5.73 × 5.73 × 19.68 Å) were constructed in an orthogonal box, including a vacuum layer of 20 Å. Moreover, the binding energies

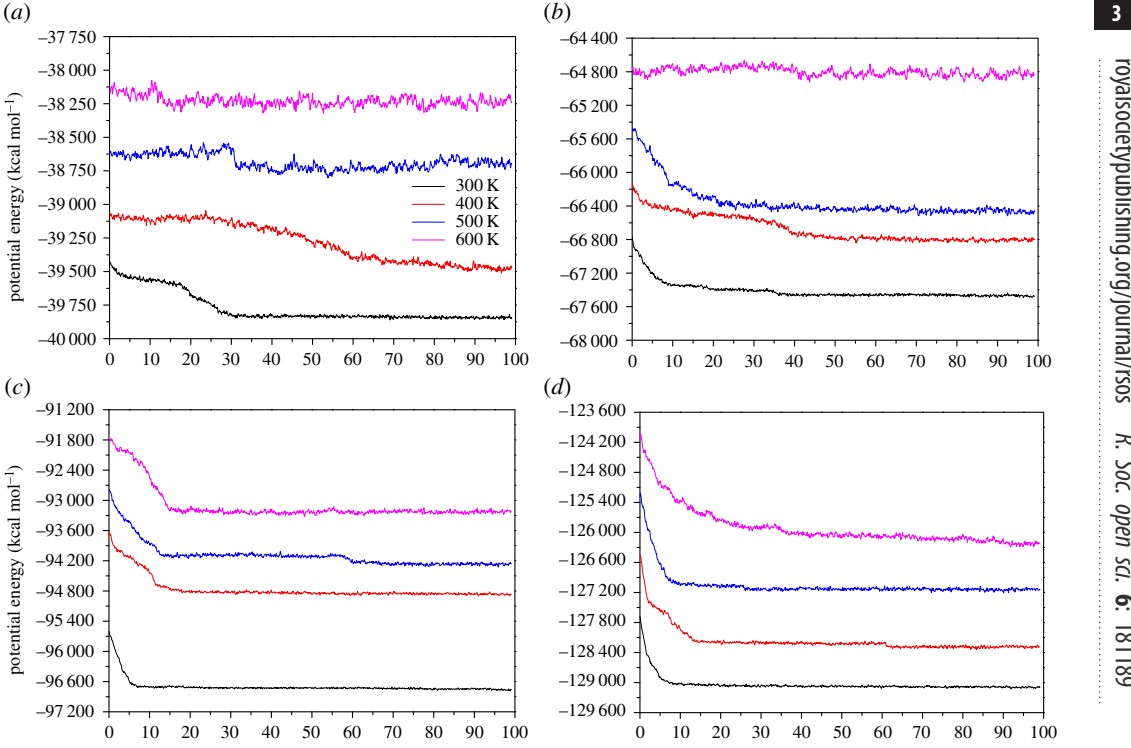

**Figure 1.** Potential energy of Al substrate with different layers: (*a*) three layers; (*b*) five layers; (*c*) seven layers; (*d*) nine layers.

of three different pairs were tested, the Al–O, Al–OH and Al–H pairs were calculated in turn and $5 \times 5 \times 1$ k-point grid was used. The results of QM calculations are discussed in 3.1.

## 2.2. Simulation system

Ethanol molecules play a crucial role in the entire process. The configuration of ethanol molecules was obtained by density functional theory (DFT) calculations. We adopted the BFGS geometry optimization method to find the most stable ethanol molecule structure [16]. The simulation box is set up as a cubic box with dimensions of $41 \times 33 \times 55$ Å and an Al slab with (1 1 1) surface was set in the region of $41 \times 33 \times 16.2$ Å. 8 Å above the metal slab is a region filled with ethanol molecules in different densities. The Berendsen thermostat method was used to control the temperature of the system with a damping constant of 50.0 fs [17]. Because our simulations were under a relatively low temperature range (200–500 K), a time step of 0.5 fs is enough to describe the adsorption behaviours between ethanol molecules and Al atoms. To guarantee every system stays in the ground state at the beginning of simulation, we performed the energy minimization process which used the conjugate gradient algorithm and the energy tolerance was set to $1.0 \times 10{-}10$ Kcal mol$^{-1}$. The equations of atomic motion are integrated by the Verlet velocity algorithm [18]. All MD simulations were carried out by LAMMPS with the USER-REAXC package. VMD and OVITO were chosen as post-processing software and visualization software, respectively [19–22]. In adsorption simulations, the thickness of the metal substrate is a major factor that may affect the final results, because the coordination numbers of the surface atoms are lower than those of the internal atoms and the binding force of surface atoms is also smaller. When the temperature is relatively high, a thinner Al substrate will melt before the beginning of the adsorption process. A too-thick substrate will significantly increase the calculation time. To determine a reasonable number of Al layers and eliminate surface melting effect, we set a series of relaxation calculations, each of which has three, five, seven or nine layers of Al atoms. Figure 1 shows the variation of system energy with different temperature in 100 ps.

The case of three Al layers shows a disordered relaxation trend: at low temperature (300 K), the total energy decreases slowly with fluctuations, but when the temperature is above 400 K the substrate melts directly which is reflected from the curves keeping almost horizontal from the beginning. The test results

of five and seven Al layers are not ideal either: the five-layer slab melted at 500 K and both energies vary with temperature nonlinearly. Only the case of nine layers showed a robust structure: the total energy did not fluctuate with the temperature obviously but rose proportionately with the temperature (average 300 kcal mol$^{-1}$ per 100 K). The above analyses indicate more than nine layers of Al substrate are thick enough to eliminate the surface effects. Therefore, the Al slab with nine layers was chosen as the substrate in subsequent simulations.

# 3. Results and discussion

## 3.1. Force field validation

The reactive force field (ReaxFF) interatomic potential is a method that combines the fitting results of quantum mechanics (QM) calculations and empirical interatomic potential within the bond-order formalism. Therefore, ReaxFF can implicitly describe chemical bonding, neither predefining the connectivity between atoms nor performing expensive QM computations. Unlike other traditional force fields which usually predefine connectivity between atoms, ReaxFF adopts bond-order formalism which is derived from Tersoff bond-order/distance relationship to judge interactions between atoms including bond and long-range pair interactions. Bond-order mechanism is the core conception of ReaxFF and equation 1 interprets its computation method in the empirical formula:

$$\text{BO}_{ij} = \text{BO}_{ij}^{\sigma} + \text{BO}_{ij}^{\pi} + \text{BO}_{ij}^{\pi\pi} = \exp\left[ p_{\text{bo1}} \left( \frac{r_{ij}}{r_0^{\sigma}} \right)^{p_{\text{bo2}}} \right]$$
$$+ \exp\left[ p_{\text{bo3}} \left( \frac{r_{ij}}{r_0^{\pi}} \right)^{p_{\text{bo4}}} \right] + \exp\left[ p_{\text{bo5}} \left( \frac{r_{ij}}{r_0^{\pi\pi}} \right)^{p_{\text{bo6}}} \right], \tag{3.1}$$

where BO is the bond order between atoms $i$ and $j$, $r_0$ terms are equilibrium bond lengths and $p_{\text{bo}}$ terms are empirical parameters. The equation is continuous and contains no discontinuities through transitions between $\sigma$, $\pi$ and $\pi\pi$ bond characters. Note that bond order does not include all pair interactions in the system. Instead, the force field will build a bonded neighbour list for every atom to avoid spurious bond characters and any excessive close-range non-bonded interactions are avoided by the inclusion of a shielding term. During molecular dynamics simulation processes, bond-order neighbours will be updated by every iteration step to recalculate all bonded interactions.

The ReaxFF force field can well describe complex chemical reactions and intra-molecular structural changes by monitoring distance. Over nearly 10 years of development, the ReaxFF method has been successfully applied in many fields such as the studies of heterogeneous catalysis, atomic layer deposition and so on [23].

In this paper, all simulations were carried out by using the ReaxFF force field which was specially developed for systems including C, H, O and Al. The origin's potential file and further details on the training processes of this force field can be obtained from [7]. More information about the ReaxFF concept can be found in the works of van Duin and co-workers [24,25].

The force field file used had been proven to be valid for a qualitative description of Al–C bonds and hydrocarbon structure in [7]. In this paper, the binding energy of parts of the hydroxyl group is of great interest to us. We compared the ReaxFF results with additional DFT calculations. The binding energy was calculated as follows:

$$E_{\text{binding energy}} = E_{\text{group1}} + E_{\text{group2}} - E_{\text{pair}}, \tag{3.2}$$

where $E_{\text{group}}$ is the energy of different parts and $E_{\text{pair}}$ is the total energy after two parts binding together. Figure 2 shows the comparison results of the binding energy obtained by ReaxFF MD simulations and DFT results. A similar energy trend indicates that the ReaxFF force field can reasonably simulate the adsorption processes of ethanol molecules on the Al surface.

## 3.2. Single-ethanol molecule adsorption mechanism

To investigate the adsorption mechanism of ethanol molecules, we carried out single-molecule adsorption simulations first. The model of ethanol molecule comes from the DFT optimization calculation. The molecule with atom number is depicted in figure 3. Bond and charge information of

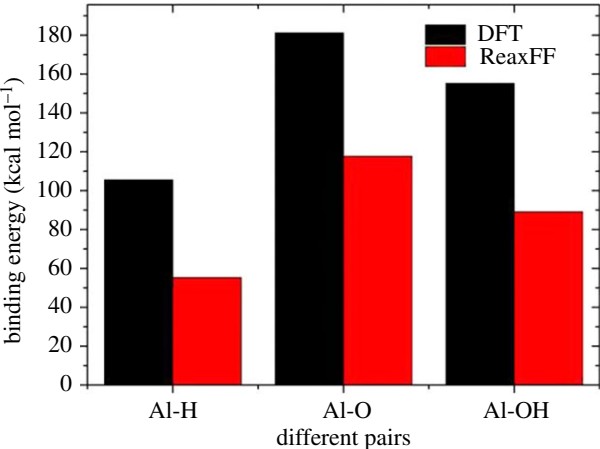

**Figure 2.** Comparison of binding energies of Al−O, Al−H and Al−OH pairs obtained by DFT and ReaxFF.

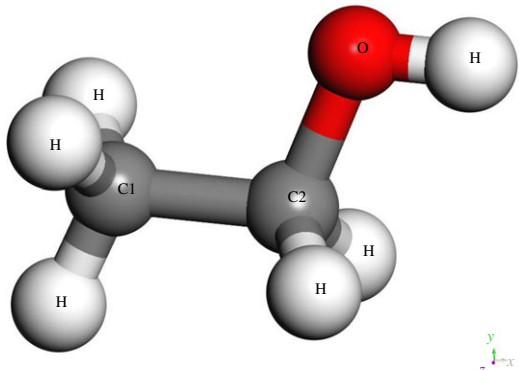

**Figure 3.** Structure configuration of the ethanol molecule with atomic numbers.

single-ethanol molecule are listed in tables 1 and 2, respectively. All single-molecule adsorption simulations were performed under 300 K to observe the most stable adsorption structure.

According to the studies of A. Korherr *et al.*, at room temperature (300 K), the thermal vibrations of substrate atoms can be neglected if one is mainly interested in the interaction with an absorbed molecule and not in the physical behaviour of the crystal itself [26]. Therefore, in single-molecule adsorption simulations, all Al atoms were fixed, but interaction forces were retained under 300 K. Time step 0.1 fs was chosen in subsequent single-molecule adsorption simulations to observe the structural evolution process in more detail. Figure 4 shows the trajectory of the single-ethanol molecule at different periods using the charge colouring method.

The adsorption process was completed in 5 ps. Initially, the ethanol molecule was placed above 5 Å from Al substrate. Affected by thermal motion, the ethanol molecule was constantly adjusted to make the hydroxyl group point toward the surface of Al substrate. At 3 ps, some local surface Al atoms began to be affected by a nearby ethanol molecule and turned positive. At 4 ps, driven by electrostatic force between the hydroxyl oxygen atom and Al atoms, the ethanol molecule headed to the surface. 5 ps and later, the ethanol molecule began to vibrate near the final position for which the average value of the distance between hydroxyl oxygen atom and surface is 2.39 Å. The ethyl group keeps rotating around the hydroxyl group. The plot of potential energy varying with time is shown in figure 5. After completion of the adsorption process, potential energy fluctuates near $-194\,085\,\text{kcal mol}^{-1}$ which can be attributed to the thermal motion of the ethanol molecule.

However, due to the asymmetry of ethanol molecules, it is hard to judge the beginning of the adsorption process by the distance between the molecule centroid and the surface, especially when the ethanol molecule is decomposed into two parts: the hydroxyl (OH) and ethyl ($C_2H_5$) group. Besides, the guiding role of hydroxyl in the adsorption process also needs to be further proven. Therefore, the adsorption processes of single ethyl and hydroxyl groups were studied in this section. The computation settings and charge distribution are the same as in the single-ethanol molecule case,

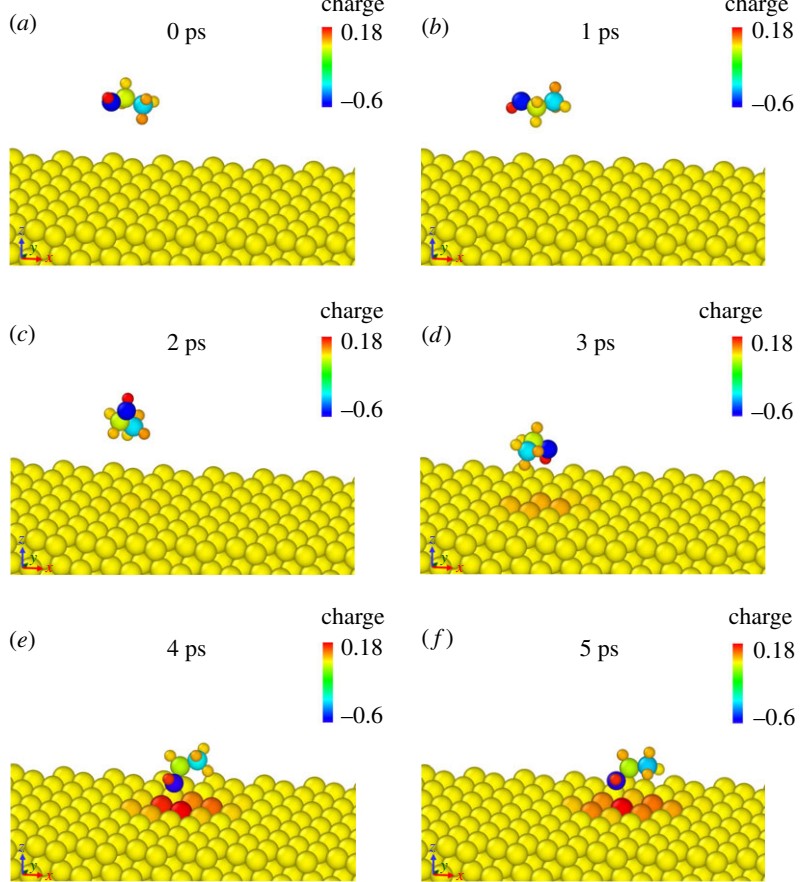

**Figure 4.** $(a-f)$ Snapshots of the single-ethanol molecule adsorption process $(0-5$ ps) coloured by charge value.

**Table 1.** Atomic effective charges of ethanol model.

| atom | $q/e$, proton charge |
|------|-----------------------|
| C1 | $-0.159$ |
| C2 | $0.054$ |
| O | $-0.57$ |
| H | $0.053$ |
| H3 | $0.41$ |

**Table 2.** The length of bonds in the ethanol model.

| bond | $R_0$ (Å) |
|------|-----------|
| C1$-$C2 | 1.523 |
| C$-$H | 1.099 |
| C$-$O | 1.439 |
| O$-$H | 0.975 |

but the absorbent was changed. Compared with the adsorption process of the single-ethanol molecule, the adsorption process of the single ethyl and hydroxyl groups is relatively simple: C2 and O atoms lead the adsorption processes of their groups, respectively. Accordingly, when calculating the adsorption distance, we regard C2 and oxygen atoms as reference atoms. Adsorption distance and potential energy analysis results are shown in figures 6 and 7.

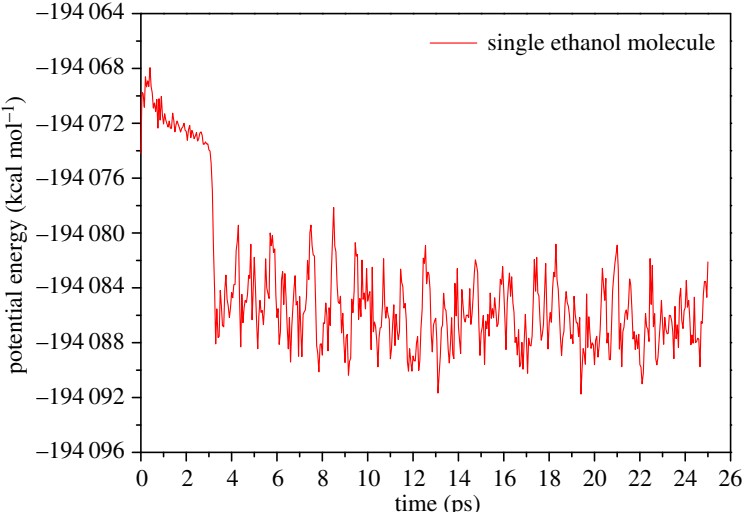

**Figure 5.** Potential energy varies with time of the single-ethanol molecule adsorption simulation.

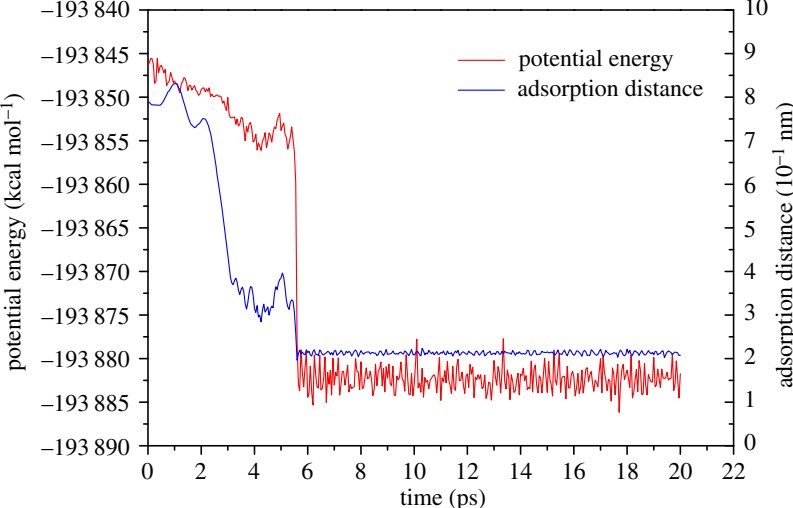

**Figure 6.** Plots of potential energy and adsorption distance vary with time for single ethyl group.

In the ethyl adsorption simulation, because there are fewer coordination numbers in the C2 atom, it is more easily attracted by surface Al atoms. As the ethyl approaches the Al surface, the valence of the C2 atom is changed to negative. Before adsorption, two peaks appear in the potential energy curve. By checking the trajectory of the ethyl group, this phenomenon is caused by the structural asymmetry of the ethyl group. At the moment of potential energy peak, the ethyl group was rotated by long-range interactions with Al atoms which makes the C2 atom head to the surface. The average adsorption distance fluctuates around 2.1 Å. Note that the trend of the ethyl potential energy curve is similar to that of the single-ethanol molecule.

For the hydroxyl group, the oxygen atom leads the adsorption process as expected. The potential energy curve decreases greatly and is horizontal after adsorption which indicates the adsorption configuration is more stable than that of ethyl groups. The hydroxyl finally adsorbs on short-bridge position 1.2 Å above the surface perpendicularly, which is lower than the ethyl group. In the process of being absorbed, the motion of the hydroxyl group is also smoother than the ethyl group. Additionally, the hydroxyl completes its adsorption process in 3 ps which is 0.5 times faster than that of single ethyl group. It is interesting that 3 ps is consistent with the adsorption time of the single-ethanol molecule which can be checked from figure 5. Such findings also provide proof of the important role of the hydroxyl group in the adsorption of ethanol molecules.

The above analyses reveal that under low temperature, physical adsorption occurs between ethanol molecules and Al substrate. The adsorption intensity and time are dependent on the performance of the

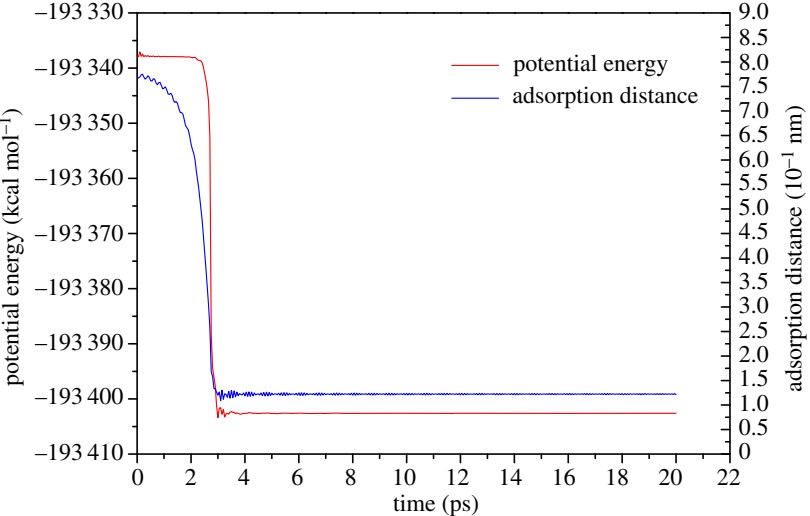

**Figure 7.** Plots of potential energy and adsorption distance vary with time for single-hydroxyl group.

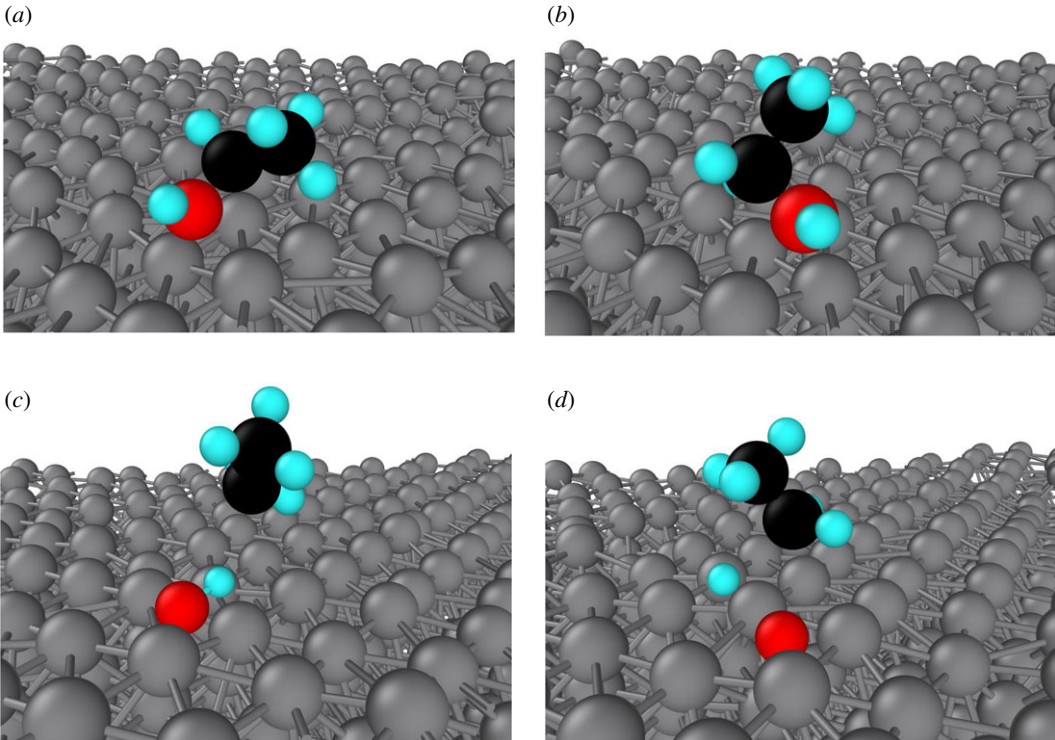

**Figure 8.** Snapshots of adsorption process of an ethanol molecule on Al surface at 400 K: (*a*) 32.5 ps; (*b*) 38.8 ps; (*c*) 61.8 ps; (*d*) 63.3 ps (grey, Al atoms; red, O atoms; black, C atoms; cyan, H atoms).

hydroxyl group and the adsorption instability is mainly responsible for the ethyl group. Although these two groups of the ethanol molecule can interact with the Al substrate, the adsorption of the hydroxyl group occurs earlier and interacts with Al atoms more firmly. Therefore, it is reasonable to regard adsorption of hydroxyl as the beginning of ethanol adsorption.

The adsorption process becomes more complicated when the temperature is so high that we have to consider the thermal vibrations. Under the influence of thermal vibration of surface Al atoms, ethanol molecules are easily decomposed into two parts: hydroxyl and ethyl groups. Hydroxyl groups prefer to diffuse into the interior region of the Al surface, but, the remaining ethyl groups keep floating on the Al surface. Figure 8 shows a series of snapshots that represent a typical example of the adsorption process under relatively high temperature (above 400 K). From the figures, we can see the surface of

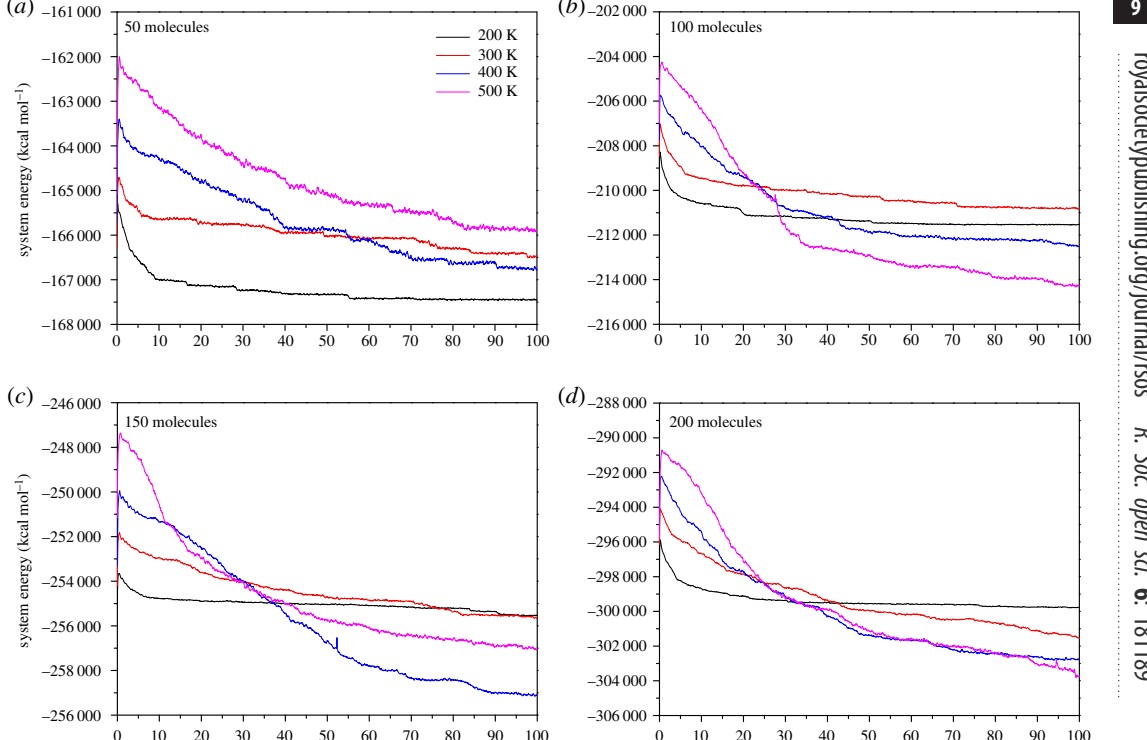

**Figure 9.** System energy versus time curves: (*a*) 50 ethanol molecules; (*b*) 100 ethanol molecules; (*c*) 150 ethanol molecules; (*d*) 200 ethanol molecules.

Al substrate becomes so rough that ethanol molecules are subjected to forces in different directions. Initially, the hydroxyl was placed parallel to the surface. After 32.5 ps, the hydroxyl begins to head to the Al surface, as shown in figure 8*a* and *b*. Subsequently, the hydroxyl departs from the molecule due to the influence of thermal vibration of surface Al atoms and electrostatic attraction. Finally, the hydrogen atom and the oxygen atom are separated and they spread to the interior of the surface (figure 8*c,d*).

Combined with all the above analyses, we summarized that the adsorption process of ethanol molecules on the Al surface is not a simple physical adsorption process but a combination of physical and chemical processes, especially when the environment is above room temperature. The adsorption process of ethanol molecules is divided into two parts: ethyl and hydroxyl groups. We categorize the adsorption process into two modes. The first adsorption mode is ethanol molecules adsorbed on the short-bridge sites of the Al surface and the other is the chemisorption mode.

## 3.3. Adsorption process

### 3.3.1. System energy and adsorption rates

We performed NVT molecular dynamic simulations for different ethanol molecules under temperatures ranging from 200 to 500 K with 100 K increments. Before the start of the adsorption processes, energy minimization processes were performed as described in §2.2. Data points were recorded every 0.05 ps. Figure 9 shows that the total energies of different systems varied with time at different temperatures. It is clear that when the temperature is 200 K, all curves descend smoothly and are converged to a certain value. Such curves indicate, at low temperature, that the system experiences a simple adsorption process in which ethanol molecules are adsorbed by Al substrate and release adsorption energy using the first adsorption mode. When the temperature reaches 300 K, the curves show different adsorption routes: for 50 and 100 molecules, the adsorption curve can still converge into a certain value, but it approaches closer to the 200 K curve over time. However, the situation becomes quite different in 150 and 200 molecules: the curves keep their downward trend from the beginning to the end and finally reach values below the endings of corresponding 200 K curves. We attribute such

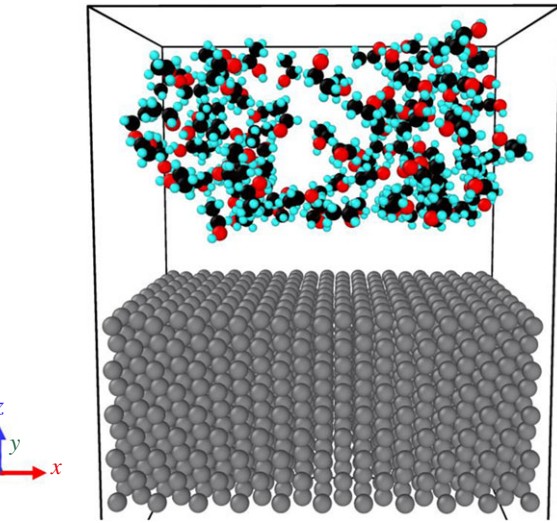

**Figure 10.** Snapshot of an ethanol–Al adsorption system at 200 K (coloured by elements).

**Table 3.** General information of ethanol adsorption simulations in different temperature and pressure.

| no. ethanol molecules | gas pressure (kPa) | simulation temperature range (K) | Al substrate atoms | total no. atoms |
|---|---|---|---|---|
| 50 | 1.16 | 200–500 | 1607 | 2057 |
| 100 | 2.33 | 200–500 | 1607 | 2507 |
| 150 | 3.49 | 200–500 | 1607 | 2957 |
| 200 | 4.65 | 200–500 | 1607 | 3407 |

a phenomenon to the increase of the number of adsorbed molecules, because our previous simulations proved no ethanol molecules decompose under 300 K and the first adsorption mode fits this situation. When temperature comes to a relatively high value (400 K and 500 K), all curves share a similar declining trend: before 30 ps, the system energy declines sharply. After that, the slope of curves approaches zero. Apart from the case of 50 ethanol molecules, the final system energy of curves reaches a value below the situations of 200 K and 300 K at 100 ps. It is expected that due to the high temperature, the second adsorption mode dominates the adsorption progress. The case with 50 ethanol molecules is limited for fewer adsorbed molecules, so the energy did not decrease a lot.

In subsequent adsorption simulations, each case consists of an Al substrate and a different number of ethanol molecules which controls the pressure in vacuum above the substrate. Information on adsorption simulation cases is listed in table 3. Figure 10 shows a snapshot of an equilibrated state of a system with $N = 100$ ethanol molecules.

To understand the adsorption process in more detail, we plot the number of adsorbed molecules as a time function in figure 11. According to the analyses in §3.2, there are two adsorption modes. When counting the number of absorbed molecules, we consider two adsorption types at the same time: firstly, we identify all the ethanol molecules with a sequence number. Then, we regard ethanol molecules with oxygen atoms which are less than 2.39 Å away from Al atoms as effective absorbed molecules. Besides, in the chemisorption mode, the adsorption of oxygen and carbon atoms occurs simultaneously. The ethyl groups shedding off the ethanol molecules are considered to be effective absorbed molecules if the distance between carbon atoms and Al atoms is within 2.1 Å. Owing to the adoption of molecular identifier, all carbon and oxygen atoms which stem from the same molecule conforming to the adsorption standard will be counted as one absorbed molecule.

We can see how temperature and pressure (controlled by the number of ethanol molecules) factors affect the adsorption process in figure 11. At the same temperature, it is clear that the final adsorption number of ethanol molecules increases with the increase of pressure and the temperature obviously improves the adsorption rate. The whole adsorption process can be divided into two periods: the first

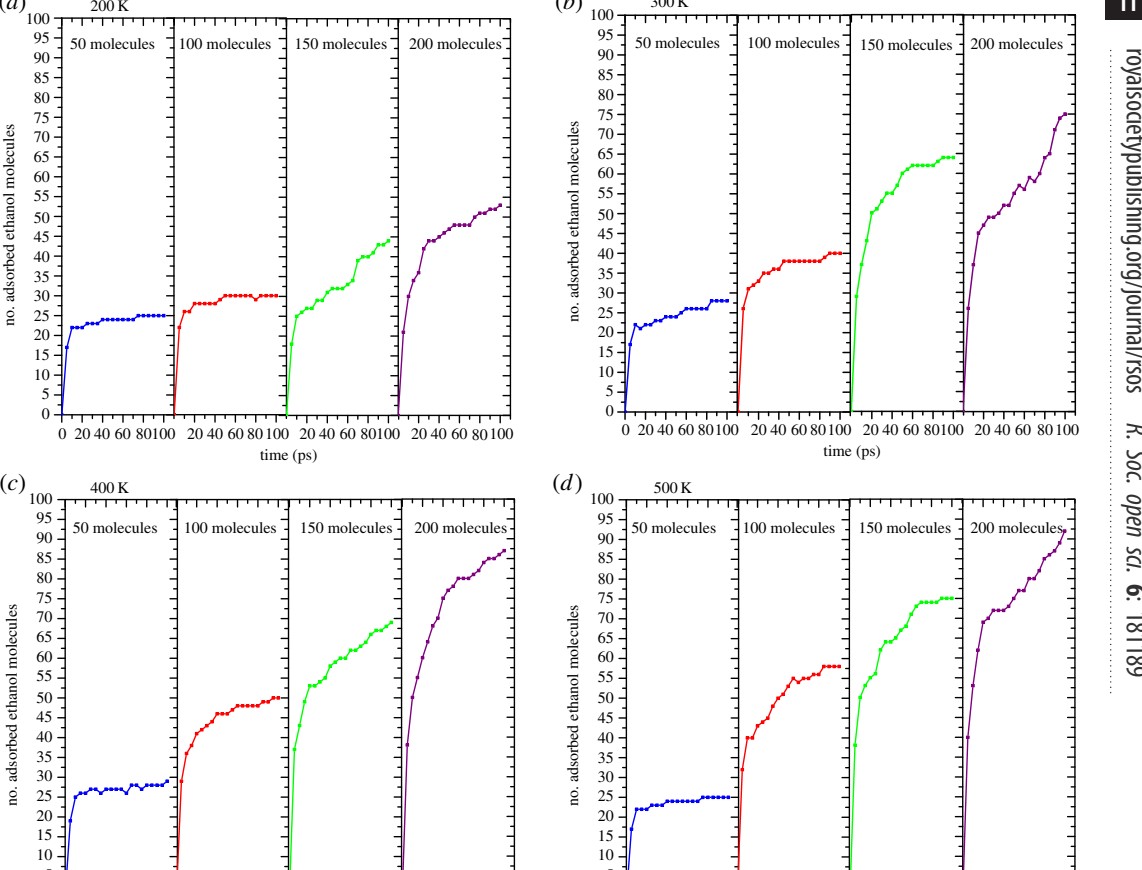

**Figure 11.** Isothermal adsorption curves for different numbers of ethanol molecules.

period (typically before 30 ps) is a process of rapid growth of adsorbed ethanol molecules. Temperature plays a crucial factor in this period. When the pressure is constant, the higher the temperature, the higher the number of ethanol molecules adsorbed in the first period. Note that for cases with less ethanol molecules (50 and 100 molecules), the height reached in the first period largely determines the final adsorption number. Because with the process of adsorption, the pressure of other ethanol molecules is decreasing which also proves the rate of adsorption and the saturated adsorption number are proportional to the pressure; the second period is characterized by a rise in fluctuations for 150 and 200 molecules. It is expected that a higher temperature makes ethanol molecules vibrate more intensely on the Al surface which gives other ethanol molecules more chance to bind to free sites on the Al surface. Though the general trend is rising, there are still some reductions in the curves which proves that the adsorption process is not a static process but a process of competition for binding sites. Unlike the first period, adsorption and desorption processes coexist in this period. We speculate that all free sites have been occupied by adsorbed ethanol molecules in the first period and they adjust themselves under different temperature to expose more binding sites. This assumption can also be proved by the continuously reduced energy plots in figure 9. However, for cases with fewer molecule numbers, their adsorption curves show a stepwise shape which reveals that the temperature is helpful for raising the quantity of adsorbed ethanol molecules, but it plays a limited role in terms of improving the adsorption rate.

### 3.3.2. Intermolecular structure

As mentioned in the §1, coating ANPs with organics is performed at room temperature. Surface properties with a coating layer at 300 K are studied in this section. The structure of liquid is usually

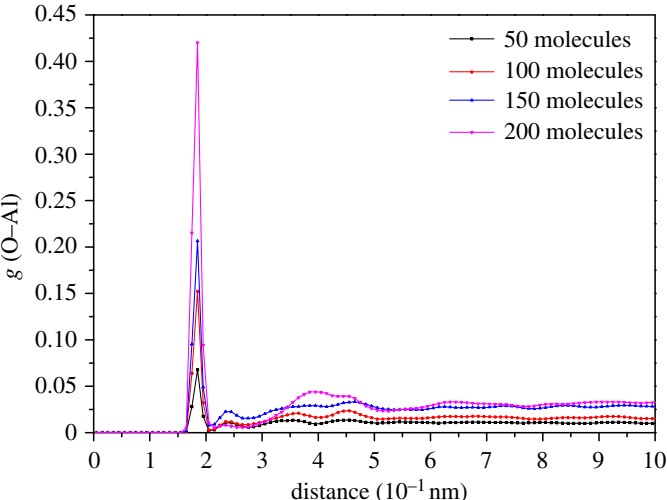

**Figure 12.** The RDF of O−Al pair of different cases at 300 K.

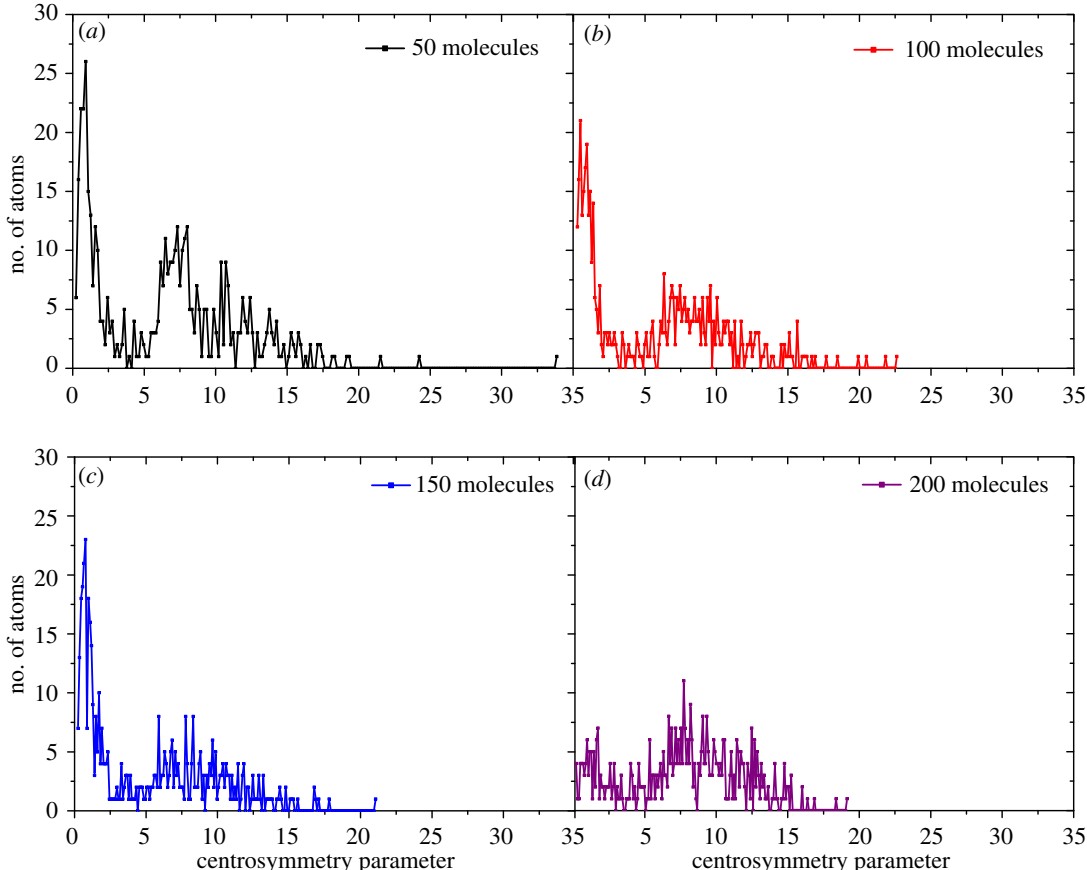

**Figure 13.** CSP analysis of Al surface under different pressures at 300 K.

expressed by radial distribution functions (RDF) g(r). All RDF results were calculated by the VMD code [20]. Equation (3.3) shows how g(r) is calculated in molecular dynamic simulations.

$$g(r) = \frac{1}{\rho 4\pi r^2 \delta r} \frac{\sum_{t=1}^{T} \sum_{j=1}^{N} \Delta N\left(r \xrightarrow{\Delta} r + dr\right)}{N \times T}, \quad (3.3)$$

where $\rho$ is the system density (quantity density), $T$ is the total computation time (steps), $N$ is the total number of atoms and $r$ is the radius away from the reference atom. The $dr$ was 0.1 Å in subsequent RDF analyses. The most interesting $g(r)$ function is the O−Al bonding because the oxygen atoms in

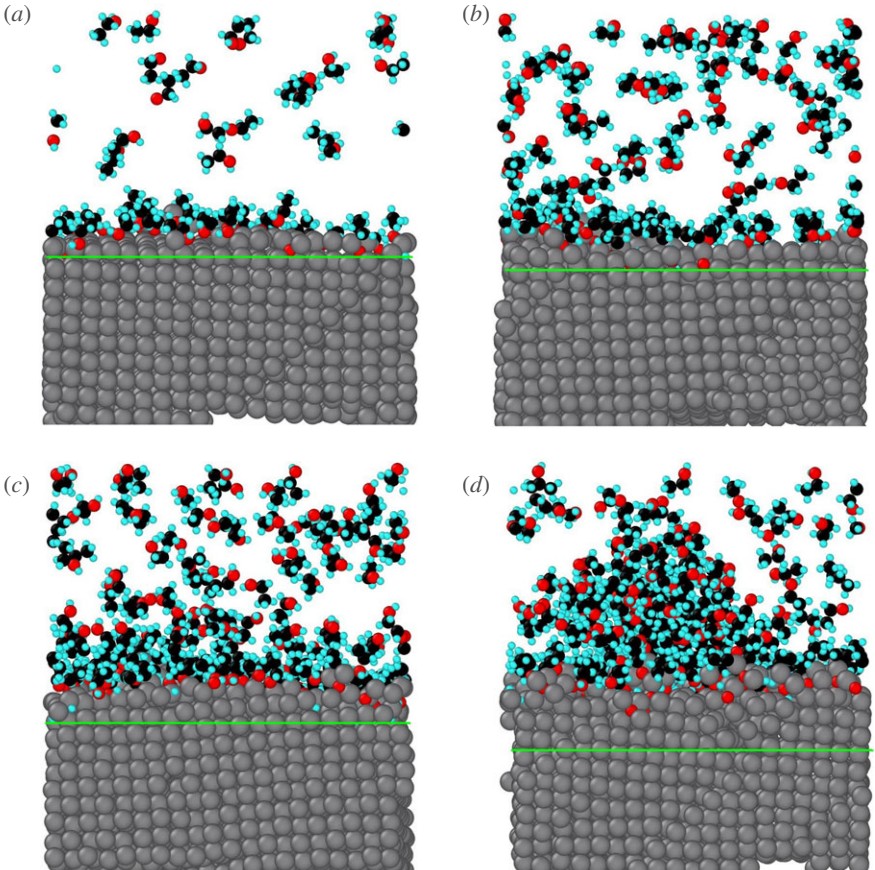

**Figure 14.** Final snapshots of Al–ethanol system with different initial ethanol molecules. The green line highlights the boundary of the region affected by atoms. (coloured by elements) (a) 50 molecules, (b) 100 molecules, (c) 150 molecules, (d) 200 molecules.

the hydro groups are the leading atoms in the adsorption processes. All RDF plots depicted in this section are computed from the last 1000 frames which represent the final configuration instead of providing initial and medium-term information on adsorption processes.

First, we studied the effect of pressure on the adsorption process at 300 K. Figure 12 shows the O–Al RDF graphs of cases with different initial ethanol molecules.

The trends of different plots are similar and it is obvious that the first peak value is proportional to the total number of ethanol molecules. This finding also provides further evidence for the previous speculation. Additionally, the position of the first peak indicates the bonding distance between the Al surface and ethanol molecules. All cases at 300 K almost share the same adsorption distance of 1.85 Å, which is less than the 2.39 Å analysed in §3.2. This indicates that in the actual adsorption process, the second adsorption mode is more popular. The second peak of 50, 100 and 150 cases appears at 2.3 Å which is identified as the distance of the first adsorption mode. However, for the case with 200 ethanol molecules, the second peak value appears around 4 Å (a distance relative far from the Al surface). It is expected that so many ethanol molecules continuously pound the surface Al atoms that the surface became too rough to provide available binding sites for the first adsorption mode.

To prove our speculation, we calculated centro-symmetry parameter (CSP) of the top four Al layers. CSP is computed by the following equation:

$$CS = \sum_{i=1}^{N/2} |\vec{R_i} + \vec{R}_{i+N/2}|^2, \tag{3.4}$$

where $N$ are the nearest neighbours of each atom identified and $R_i$ and $R_{i+N/2}$ are the vectors from the central atom to a particular pair of the nearest neighbours. In solid-state systems, the CSP is a useful measure of the local lattice disorder around an atom and can be used to characterize whether the atom is part of a perfect lattice, a local defect or at a surface. For fcc lattices, when the value of CSP is 0, it means that the atom is surrounded by atoms on a perfect lattice and the larger the value, the more the structure deviates from the

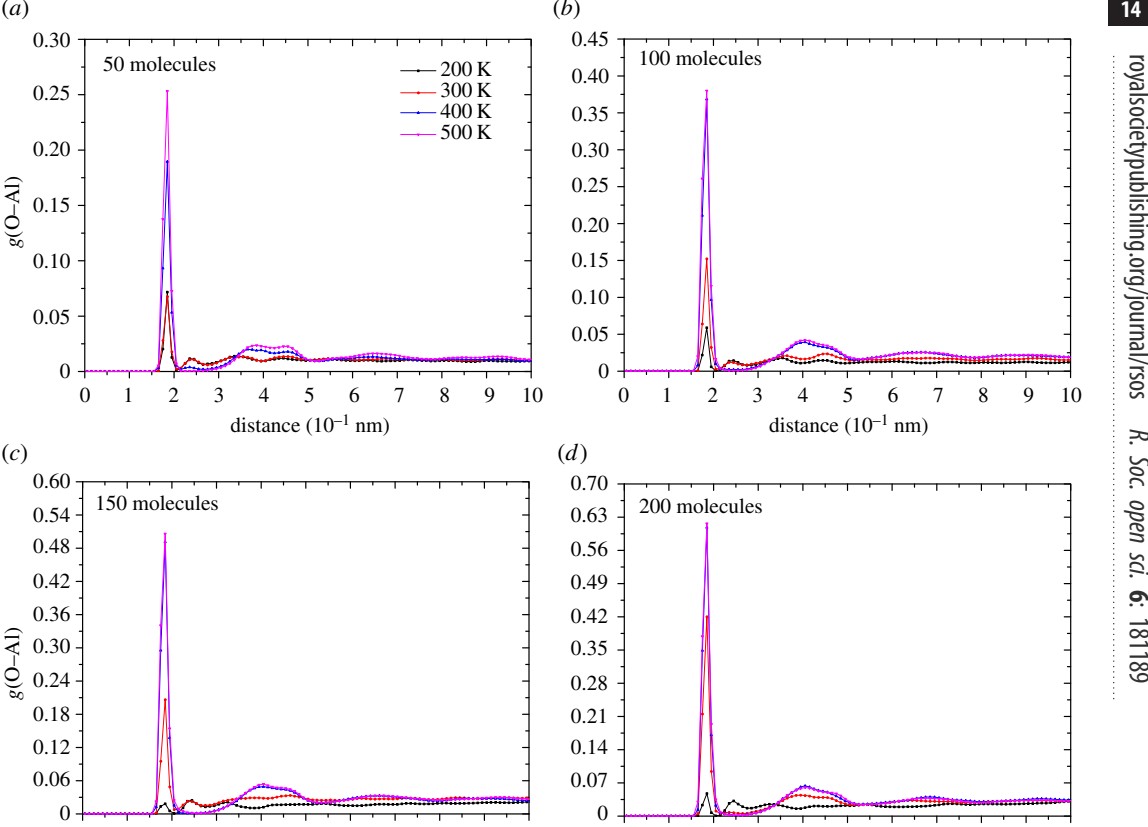

**Figure 15.** RDF plots of O−Al pair varies with different temperatures.

perfect fcc structure. Figure 13 shows the results of CSP calculations. It is clear that the distribution trends of cases with less than 200 ethanol molecules are similar: they all have a peak within the range of 0−5 Å which indicates that they still maintain a relatively complete crystal structure on their surface. However, the situation is quite different for the case with 200 ethanol molecules: the CSP value of surface atoms distributes more uniformly and one obvious peak appears around 8 Å which cannot be considered as a crystal state. Based on the above analysis, our assumption is reasonable.

We collected final configurations of the system with different initial ethanol molecules in figure 15 and drew mark lines to show the region affected by adsorption in the Al slab to prove this point further. It is obvious that surface Al atoms in figure 14d have been totally disordered by adsorbed ethanol molecules and oxygen atoms diffuse into a deeper region than other cases which corresponds with findings from figure 13. Additionally, we studied how the temperature affects the adsorption structure and RDF results can be seen in figure 15.

All cases show a similar trend of RDF with temperature. The first peak value indicates ethanol molecules adsorbed by the second adsorption mode which obviously increases with temperature. The rise of temperature does not move plots horizontally but elevates plots overall. Therefore, increasing the temperature will also have a positive effect on increasing the number of adsorbed ethanol molecules, especially when the pressure is sufficient. Note that the ratio of the maximum and minimum peaks in one chart differs greatly. Under lower pressures, the ratio is limited within 10, however, for higher pressures the ratio can reach hundred of orders of magnitude. Such a phenomenon indicates that the pressure greatly increases the probability of ethanol molecules appearing around the surface.

## 4. Conclusion

In this paper, we performed molecular dynamic methods with the ReaxFF force field to study the process of Al slabs adsorbed by ethanol molecules under canonical ensemble. Simulations with a time step of 0.1 fs show that the hydroxyl group plays an important role in the adsorption process of ethanol molecules. The electrostatic force between O and Al atoms is the most stable in the whole ethanol molecule.

However, the ethyl group is responsible for the instability in adsorption processes due to its structural asymmetry. The adsorption distances for ethyl and hydroxyl groups on the Al surface are 2.1 and 1.2 Å, respectively. The adsorption between ethanol molecules and Al is a combination of physical and chemical processes which is more popular under relatively high temperatures. The whole adsorption process can be divided into two periods characterized by adsorption rates. For 50 and 100 cases, the first period greatly determines the final number of adsorbed molecules and the second period prefers to develop a step form rise. For higher pressure cases, the second period reflects that molecules are adsorbed on the surface continually and compete for free-binding sites from energy analyses and isotherm adsorption curves. CSP analysis reveals that when the number of ethanol molecules reaches 200, the surface Al atoms become so rough that they cannot maintain a complete lattice structure. O−Al RDF was carried out from 200 K to 500 K and the results show that raising the temperature can help improve the final adsorption quantity, especially under sufficient pressure.

Data accessibility. The datasets supporting this article have been uploaded as part of the Recommendation File.

Authors' contributions. P. L. is the fund applicant and provided the idea for this paper. J. L. collected and analysed the data and wrote the manuscript. M. W. provided language guidance. All authors gave final approval for publication.

Competing interests. We declare we have no competing interests.

Funding. This work was supported by the Fundamental Research Funds of Harbin Engineering University of China (no. HEUCFP201780).

Acknowledgements. We are grateful for the necessary support from the Fundamental Research Funds of Harbin Engineering University of China (no. HEUCFP201780).

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
