## [Reviewer comments · Royal Society Open Science]

Review History

RSOS-181189.R0 (Original submission)

Review form: Reviewer 1 (Kai Luo)

Is the manuscript scientifically sound in its present form?

Yes

Are the interpretations and conclusions justified by the results?

No

Is the language acceptable?

No

Is it clear how to access all supporting data?

Yes

Do you have any ethical concerns with this paper?

No

Have you any concerns about statistical analyses in this paper?

No

Recommendation?

Reject

Comments to the Author(s)

This study investigated the adsorption process of ethanol molecules on Al surface using ReaxFF MD simulations. Two adsorption modes were identified and the effects of different temperatures and pressures on the adsorption process were studied. The results of system energy and RDF were also presented to describe the adsorption process. Overall, the results are interesting but the importance of the findings was not established and the scope and depth of the study are very limited. Moreover, several issues need to be addressed.

1. The English of this manuscript needs to be improved. There are numerous minor grammatical as well as some typo errors (e.g. Page 6, Line 23: Am; Page 13, Line 51: Struvture; Page 16, Line 27: 'Show a similar trend that changes with temperature?'). Also, the presentation of this manuscript is poor (e.g. the font size of all the figures is not consistent; the font size changes from Page 13, Line 58; Page 6, Line 46: '3.1 section' should be '4.1 section'; Fig. 6 does not show which subfigure refers to which temperature). The manuscript should be carefully checked before submission.
2. The authors used different numbers of ethanol molecules to control the pressure. Please specify the exact initial pressure value for each simulation case (i.e. 50, 100, 150 and 200 ethanol molecules at 200 K).
3. The algorithm for calculation of adsorbed molecules could be optimized. For the first adsorption mode, although the authors mentioned that the oxygen atom always heads to the surface Al atoms in the single-molecule adsorption case, other adsorptions such as Al-H could also happen for multi-molecule adsorption. For the second adsorption mode, the authors used the subtraction of total molecule number and number of ethanol molecules outside the adsorption region as the number of adsorbed molecules. However, this number includes the decomposed ethanol molecules, which should not be counted as the adsorbed ethanol molecules. Therefore, a better method to calculate the number of adsorbed ethanol molecules may be developed.
4. Page 11, Line 51-53: 'In the end, cases 65 ethanol molecules at most'. It is not clear why the maximum number of adsorbed ethanol molecules is 65. Based on Fig. 6b-6d (200 molecules case), the number of adsorbed ethanol molecules could continue to increase provided that the simulation time is long enough.
5. The authors mentioned in Section 3.3 (Page 6, Line 54) 'a timestep of 0.5 fs is enough', which indicates that the 0.5 fs timestep was used for all the MD simulations. But in Conclusion (Page 16, Line 36-37), the authors stated 'Simulation with timestep 0.1 fs'. Please clarify this.
6. The authors validated the ReaxFF force field used in this study by comparing its results of binding energies with DFT calculations. But is there any experimental evidence that can support the ReaxFF MD simulation results obtained from this research?

7. The adsorption is said to consist of three stages. In reality, the first two stages may not be distinguishable. The third stage is said to be zero. What does it mean?

8. Temperature is said to enhance adsorption (only) but temperature may also promote desorption too. Can the conclusion be of any general value, especially when the temperature range investigated is very limited.

Review form: Reviewer 2

Is the manuscript scientifically sound in its present form?

Yes

Are the interpretations and conclusions justified by the results?

Yes

Is the language acceptable?

No

Is it clear how to access all supporting data?

Not Applicable

Do you have any ethical concerns with this paper?

No

Have you any concerns about statistical analyses in this paper?

No

Recommendation?

Major revision is needed (please make suggestions in comments)

Comments to the Author(s)

This manuscript has a clear research topic and is suitable for Royal Society Open Science. Although the research is a little common and lack of innovation in some degree, the authors combine QM calculations and MD simulations in the process of ethanol's adsorption onto Al (111) under different temperatures and pressures. The employed methods and data analysis are both reasonable and scientific.

However, the pictures in this manuscript still have large space to be made better, such as Figure 1, the degree scales on vertical axis are too dense and small, same problems are also in Figure 5 and 8. The quality of pictures in Figure 5 and 8 is poor, the readers even cannot identify the details. Additionally, in Figure 10, there are no (a)-(d) in your RDF plots corresponding to figure caption. The other colored pictures, such as Figure 2 and 9 still need to be improved.

Overall, this work has scientific analysis and reasonable explanation for simulation phenomena, but some issues are still in this manuscript, a major revision is necessary before reconsideration.

Some comments in details:

1. Your conclusion did not reflect your research result, in fact, your conclusion is only a copy of your abstract, especially the first half of Section Conclusion.
2. In the caption of Figure 8, I think that "different ethanol molecules at 300K" should be "different numbers of ethanol molecule". Similar in caption of Figure 6, "Al slab at different

temperature” should be “Al slab at different temperatures”, more important, what are different temperatures? Why not notice temperature values in figure caption? This kind of inappropriate description frequently appear in the manuscript, the authors should pay more attentions on English writing.

3. On page 13, line 51, the subheading “4.2.2 intermolecular struvture”, I think should be “intermolecular structure”, this kind of careless errors need to be avoided in the writing, especially in scientific manuscripts. The similar problems should be revised by the authors.

4. In Section 4.2.2, the authors mentioned “the most interesting $g(r)$ functions is the O-Al bonding”, in molecular dynamics simulations, the molecular and atomic interactions are common, RDF reflect the structure of atom (group) around one specific atom (group). However, how the authors can define the O-Al bonding in the adsorption of ethanol molecules on Al (111)? Can the authors explain it? Maybe under high temperature and pressure, there will be some chemical bonding forming. There is an obvious difference between physical interaction and chemical bonding.

5. In the description of Figure 6, when the number of ethanol molecule is 150, there is an obvious decreasing trend of adsorbed molecule number from 300K to 400K, while in cases of containing 200 ethanol molecules at 300K, 400K or 500K, the final adsorption number reach 65. How do the authors explain this abnormal phenomenon?

6. which indicates the Al slab used in this paper that can adsorb 65 ethanol molecules at most.

7. The cited references are old and less. Some important recent references need to be added. As far as we know, some related work about adsorption of single/multiple molecule(s) onto two-dimensional materials can be referred in your introduction or corresponding section, such as, (1) Phys. Chem. Chem. Phys., 2017, 19(43): 29222-29231; (2) Carbon, 2014, 78: 500-509; (3) J Phys Chem Lett, 2018, 9(15): 4396-4400; (4) The Journal of Physical Chemistry C, 2018, 122(21): 11385-11391.

8. The format of cited references in your manuscript is inconsistent, some titles (Ref. 3-6) are capital at the beginning of a sentence, but some titles (Ref. 7, 10, 11, 14, 16) are capital letter in every word or for the whole of title.

Decision letter (RSOS-181189.R0)

18-Sep-2018

Dear Dr Liu:

Title: Adsorption of Ethanol Molecules on the Al (1 1 1) Surface: A molecular dynamic study
Manuscript ID: RSOS-181189

The editor assigned to your manuscript has now received comments from reviewers. We would like you to revise your paper in accordance with the referee and Subject Editor suggestions which can be found below (not including confidential reports to the Editor). Please note this decision does not guarantee eventual acceptance.

Please submit your revised paper before 11-Oct-2018. Please note that the revision deadline will expire at 00.00am on this date. If we do not hear from you within this time then it will be assumed that the paper has been withdrawn. In exceptional circumstances, extensions may be possible if agreed with the Editorial Office in advance. We do not allow multiple rounds of

revision so we urge you to make every effort to fully address all of the comments at this stage. If deemed necessary by the Editors, your manuscript will be sent back to one or more of the original reviewers for assessment. If the original reviewers are not available we may invite new reviewers.

Yours sincerely,
 Dr Laura Smith, MRSC
 Publishing Editor, Journals
 Royal Society of Chemistry,
 Thomas Graham House,
 Science Park, Milton Road,
 Cambridge, CB4 0WF, UK

Royal Society Open Science - Chemistry Editorial Office

On behalf of the Subject Editor Professor Anthony Stace and the Associate Editor Professor Hazel Cox.

RSC Associate Editor:
 Comments to the Author:
 (There are no comments.)

RSC Subject Editor:
 Comments to the Author:
 (There are no comments.)

Reviewers' Comments to Author:
 Reviewer: 1

Comments to the Author(s)

This study investigated the adsorption process of ethanol molecules on Al surface using ReaxFF MD simulations. Two adsorption modes were identified and the effects of different temperatures and pressures on the adsorption process were studied. The results of system energy and RDF were also presented to describe the adsorption process. Overall, the results are interesting but the

importance of the findings was not established and the scope and depth of the study are very limited. Moreover, several issues need to be addressed.

1. The English of this manuscript needs to be improved. There are numerous minor grammatical as well as some typo errors (e.g. Page 6, Line 23: Am; Page 13, Line 51: Struvture; Page 16, Line 27: 'Show a similar trend that changes with temperature?'). Also, the presentation of this manuscript is poor (e.g. the font size of all the figures is not consistent; the font size changes from Page 13, Line 58; Page 6, Line 46: '3.1 section' should be '4.1 section'; Fig. 6 does not show which subfigure refers to which temperature). The manuscript should be carefully checked before submission.

2. The authors used different numbers of ethanol molecules to control the pressure. Please specify the exact initial pressure value for each simulation case (i.e. 50, 100, 150 and 200 ethanol molecules at 200 K).

3. The algorithm for calculation of adsorbed molecules could be optimized. For the first adsorption mode, although the authors mentioned that the oxygen atom always heads to the surface Al atoms in the single-molecule adsorption case, other adsorptions such as Al-H could also happen for multi-molecule adsorption. For the second adsorption mode, the authors used the subtraction of total molecule number and number of ethanol molecules outside the adsorption region as the number of adsorbed molecules. However, this number includes the decomposed ethanol molecules, which should not be counted as the adsorbed ethanol molecules. Therefore, a better method to calculate the number of adsorbed ethanol molecules may be developed.

4. Page 11, Line 51-53: 'In the end, cases 65 ethanol molecules at most'. It is not clear why the maximum number of adsorbed ethanol molecules is 65. Based on Fig. 6b-6d (200 molecules case), the number of adsorbed ethanol molecules could continue to increase provided that the simulation time is long enough.

5. The authors mentioned in Section 3.3 (Page 6, Line 54) 'a timestep of 0.5 fs is enough', which indicates that the 0.5 fs timestep was used for all the MD simulations. But in Conclusion (Page 16, Line 36-37), the authors stated 'Simulation with timestep 0.1 fs'. Please clarify this.

6. The authors validated the ReaxFF force field used in this study by comparing its results of binding energies with DFT calculations. But is there any experimental evidence that can support the ReaxFF MD simulation results obtained from this research?

7. The adsorption is said to consist of three stages. In reality, the first two stages may not be distinguishable. The third stage is said to be zero. What does it mean?

8. Temperature is said to enhance adsorption (only) but temperature may also promote desorption too. Can the conclusion be of any general value, especially when the temperature range investigated is very limited.

Reviewer: 2

Comments to the Author(s)

This manuscript has a clear research topic and is suitable for Royal Society Open Science. Although the research is a little common and lack of innovation in some degree, the authors combine QM calculations and MD simulations in the process of ethanol's adsorption onto Al

(111) under different temperatures and pressures. The employed methods and data analysis are both reasonable and scientific.

However, the pictures in this manuscript still have large space to be made better, such as Figure 1, the degree scales on vertical axis are too dense and small, same problems are also in Figure 5 and 8. The quality of pictures in Figure 5 and 8 is poor, the readers even cannot identify the details. Additionally, in Figure 10, there are no (a)-(d) in your RDF plots corresponding to figure caption. The other colored pictures, such as Figure 2 and 9 still need to be improved.

Overall, this work has scientific analysis and reasonable explanation for simulation phenomena, but some issues are still in this manuscript, a major revision is necessary before reconsideration.

Some comments in details:

1. Your conclusion did not reflect your research result, in fact, your conclusion is only a copy of your abstract, especially the first half of Section Conclusion.
2. In the caption of Figure 8, I think that "different ethanol molecules at 300K" should be "different numbers of ethanol molecule". Similar in caption of Figure 6, "Al slab at different temperature" should be "Al slab at different temperatures", more important, what are different temperatures? Why not notice temperature values in figure caption? This kind of inappropriate description frequently appear in the manuscript, the authors should pay more attentions on English writing.
3. On page 13, line 51, the subheading "4.2.2 intermolecular struvture", I think should be "intermolecular structure", this kind of careless errors need to be avoided in the writing, especially in scientific manuscripts. The similar problems should be revised by the authors.
4. In Section 4.2.2, the authors mentioned "the most interesting $g(r)$ functions is the O-Al bonding", in molecular dynamics simulations, the molecular and atomic interactions are common, RDF reflect the structure of atom (group) around one specific atom (group). However, how the authors can define the O-Al bonding in the adsorption of ethanol molecules on Al (111)? Can the authors explain it? Maybe under high temperature and pressure, there will be some chemical bonding forming. There is an obvious difference between physical interaction and chemical bonding.
5. In the description of Figure 6, when the number of ethanol molecule is 150, there is an obvious decreasing trend of adsorbed molecule number from 300K to 400K, while in cases of containing 200 ethanol molecules at 300K, 400K or 500K, the final adsorption number reach 65. How do the authors explain this abnormal phenomenon?
6. which indicates the Al slab used in this paper that can adsorb 65 ethanol molecules at most.
7. The cited references are old and less. Some important recent references need to be added. As far as we know, some related work about adsorption of single/multiple molecule(s) onto two-dimensional materials can be referred in your introduction or corresponding section, such as, (1) Phys. Chem. Chem. Phys., 2017, 19(43): 29222-29231; (2) Carbon, 2014, 78: 500-509; (3) J Phys Chem Lett, 2018, 9(15): 4396-4400; (4) The Journal of Physical Chemistry C, 2018, 122(21): 11385-11391.
8. The format of cited references in your manuscript is inconsistent, some titles (Ref. 3-6) are capital at the beginning of a sentence, but some titles (Ref. 7, 10, 11, 14, 16) are capital letter in every word or for the whole of title.

Author's Response to Decision Letter for (RSOS-181189.R0)

See Appendix A.

RSOS-181189.R1 (Revision)

Review form: Reviewer 1 (Kai Luo)

Is the manuscript scientifically sound in its present form?

Yes

Are the interpretations and conclusions justified by the results?

Yes

Is the language acceptable?

Yes

Is it clear how to access all supporting data?

Yes

Do you have any ethical concerns with this paper?

No

Have you any concerns about statistical analyses in this paper?

No

Recommendation?

Accept as is

Comments to the Author(s)

The revision is satisfactory and the paper is acceptable for publication after minor corrections. The detailed comments are:

Q1: The English and presentation of this manuscript have been improved. But there are still many errors. For example, Page 1, Line 48: keep them (their) chemical activity; Page 1, Line 50: Al₂O₃, 2 & 3 should be subscripts; Page 2, Lines 15-16: there are still lack studies interpret interactions; Page, Line 17: 'detail' should be 'detailed'; Page 2, Line 27: We put our research focus on; Page 5, Line 34: are not ideal too (either); Page 5, Line 53: r₀, p_{bo}, 0 & b_o should be subscripts; Page 6, Line 16: E_{group}, E_{pair}, group & pair should be subscripts; Page 8, Line 21: 'begins' should be 'began'; Page 9, Line 30: there were two peaks appear; Page 18, Line 4: R_{i+N/2}, i_{+N/2} should be subscript. I would suggest the authors do a thorough check and correct these errors.

Q2: Table 1 shows 'Atomic effective charges of ethanol model'. Where is the initial pressure information?

Q3: Revision accepted.

Q4: Revision accepted.

Q5: Revision accepted.

Q6: Revision accepted.

Q7: It was your question. I think the revision is fine.

Q8: It was your question. I think the revision is fine.

Professor Kai H. Luo PhD (Cantab)
CEng FCI FIMechE FInstP AFAIAA FASME
UCL Mechanical Engineering
University College London
Torrington Place
London WC1E 7JE, UK

Review form: Reviewer 2

Is the manuscript scientifically sound in its present form?

Yes

Are the interpretations and conclusions justified by the results?

Yes

Is the language acceptable?

Yes

Is it clear how to access all supporting data?

Yes

Do you have any ethical concerns with this paper?

No

Have you any concerns about statistical analyses in this paper?

No

Recommendation?

Accept as is

Comments to the Author(s)

No further comments

Decision letter (RSOS-181189.R1)

19-Nov-2018

Dear Dr Liu:

Title: Adsorption of Ethanol Molecules on the Al (1 1 1) Surface: A molecular dynamic study
Manuscript ID: RSOS-181189.R1

Thank you for submitting the above manuscript to Royal Society Open Science. On behalf of the Editors and the Royal Society of Chemistry, I am pleased to inform you that your manuscript will

be accepted for publication in Royal Society Open Science subject to minor revision in accordance with the referee suggestions. Please find the reviewers' comments at the end of this email.

The reviewers and handling editors have recommended publication, but also suggest some minor revisions to your manuscript. Therefore, I invite you to respond to the comments and revise your manuscript.

Because the schedule for publication is very tight, it is a condition of publication that you submit the revised version of your manuscript before 28-Nov-2018. Please note that the revision deadline will expire at 00.00am on this date. If you do not think you will be able to meet this date please let me know immediately.

Best wishes,

Dr Laura Smith
Publishing Editor, Journals

On behalf of the Subject Editor Professor Anthony Stace and the Associate Editor Professor Hazel Cox.

RSC Associate Editor:
Comments to the Author:
(There are no comments.)

RSC Subject Editor:
Comments to the Author:
(There are no comments.)

Reviewer comments to Author:
Reviewer: 2

Comments to the Author(s)
No further comments

Reviewer: 1

Comments to the Author(s)
The revision is satisfactory and the paper is acceptable for publication after minor corrections.
The detailed comments are:

Q1: The English and presentation of this manuscript have been improved. But there are still many errors. For example, Page 1, Line 48: keep them (their) chemical activity; Page 1, Line 50: Al₂O₃, 2 & 3 should be subscripts; Page 2, Lines 15-16: there are still lack studies interpret interactions; Page, Line 17: 'detail' should be 'detailed'; Page 2, Line 27: We put our research focus on; Page 5, Line 34: are not ideal too (either); Page 5, Line 53: r₀, p₀, 0 & b₀ should be subscripts; Page 6, Line 16: E_{group}, E_{pair}, group & pair should be subscripts; Page 8, Line 21: 'begins' should be 'began'; Page 9, Line 30: there were two peaks appear; Page 18, Line 4: R_i+N/2, i+N/2 should be subscript. I would suggest the authors do a thorough check and correct these errors.

Q2: Table 1 shows 'Atomic effective charges of ethanol model'. Where is the initial pressure information?

Q3: Revision accepted.

Q4: Revision accepted.

Q5: Revision accepted.

Q6: Revision accepted.

Q7: It was your question. I think the revision is fine.

Q8: It was your question. I think the revision is fine.

Professor Kai H. Luo PhD (Cantab)
CEng FCI FIMechE FInstP AFAIAA FASME
UCL Mechanical Engineering
University College London
Torrington Place
London WC1E 7JE, UK

Author's Response to Decision Letter for (RSOS-181189.R1)

See Appendix B.

Decision letter (RSOS-181189.R2)

03-Dec-2018

Dear Dr Liu:

Title: Adsorption of Ethanol Molecules on the Al (1 1 1) Surface: A molecular dynamic study
Manuscript ID: RSOS-181189.R2

It is a pleasure to accept your manuscript in its current form for publication in Royal Society Open Science. The chemistry content of Royal Society Open Science is published in collaboration with the Royal Society of Chemistry.

On behalf of the Subject Editor Professor Anthony Stace and the Associate Editor Professor Hazel Cox.

RSC Associate Editor
Comments to the Author:
(There are no comments.)

Reviewer(s)' Comments to Author:

Appendix A

Response to Reviewers

Dear editor, we have answered reviewer's comments item by item. Both the content and the format of this paper have been greatly improved. The general improved content information is listed as follows:

--- 4.2 section was added to study the mechanism of single ethanol molecule adsorption which also includes simulations for the ethyl and hydroxyl groups.

--- 11 new figures and 3 new tables were added.

--- The method for counting adsorbed molecules was improved to be more scientific. The detail discussion can be found in the 14th page.

--- The conclusion was also modified to adapt the new content.

We expected the paper can reach the publish criterion this time. Here are our detail responses:

To Reviewer 1:

1) Language was improved to ensure no minor mistakes appear; All figures in the revised paper were reproduced for better performance.

2) Initial pressure information was listed as Table 1.

3) The algorithm for calculation of adsorbed molecules has been optimized. Based on analyses of section 4.2, we developed a new method to consider physical adsorption and chemisorption at the same time. We also assigned ID numbers for each ethanol molecule to make the method more reasonable.

4) It is our fault to present such statement. Actually, after updating the counting method, the analysis process and relevant figures have changed. Detail information can be found in the Figure 11.

5) This sentence has been replaced by “Simulations with timestep 0.1 fs shows that the hydroxyl group plays an important role in the adsorption process of ethanol molecules.” In the revised paper, there are two kinds of simulations which were interpreted in section 4.2 and 4.3 respectively.

6) The ReaxFF force field used in this paper was specially trained for Al-C-H-O-N system and comparison of its results with experiments can be found in Ref 7.

7) Due to the fact that we have changed counting method, the conclusion was also changed a little. After our analysis and judgement, the adsorption process consisted of two periods now.

8) We have changed our way of expression. The temperature and pressure are two interrelated factors exerting influence on the adsorption process. Because of the new counting method, physical adsorption and chemisorption were considered at the same time which can detect the occurrence of any kind of adsorption phenomenon.

As mentioned in introduction section, we focus on process of ANPs coated by ethanol at room temperature and preserved under relatively low temperatures. Properties of coating ANPs under high temperature and reactions with oxygen will be studied in further researches.

To Reviewer 2:

1) The conclusion section has been rewritten completely according to

the new content we added and modified in the revised paper.

2) All figures have been reproduced for better performance and corresponding subtitles have been improved too.

3) The subtitle of section 4.3.2 has been replaced by “Intermolecular Structure”. We have also checked the spelling of text carefully to ensure such faults appear in the revised paper.

4) All RDF results were calculated by VMD code and calculation settings were shown in section 4.3.2.

5) After we adopted the new counting method, our findings and conclusion were also changed. The new counting method was discussed detailly in section 4.2 which can detect physical adsorption and chemisorption at the same time.

6) Maximum number of adsorbed molecules is not within the scope of this study. We focus on the adsorption mechanism of ethanol molecules and its adsorption characteristics under low temperatures. Such expressions have been deleted in the revised paper.

7) Thanks for your reference recommendations, it really helps us a lot. We carried out charge analyses like those references and make the revised paper more scientific. All references recommended were inserted in introduction section.

8) All references mentioned have been retyped and the format of references is unified.

-----Initial Editor's/Reviewers' comments-----

Reviewer: 1

Comments to the Author(s)

This study investigated the adsorption process of ethanol molecules on Al surface using ReaxFF MD simulations. Two adsorption modes were identified and the effects of different temperatures and pressures on the adsorption process were studied. The results of system energy and RDF were also presented to describe the adsorption process. Overall, the results are interesting but the importance of the findings was not established and the scope and depth of the study are very limited. Moreover, several issues need to be addressed.

1. The English of this manuscript needs to be improved. There are numerous minor grammatical as well as some typo errors (e.g. Page 6, Line 23: Am; Page 13, Line 51: Struvture; Page 16, Line 27: 'Show a similar trend that changes with temperature'?). Also, the presentation of this manuscript is poor (e.g. the font size of all the figures is not consistent; the font size changes from Page 13, Line 58; Page 6, Line 46: '3.1 section' should be '4.1 section'; Fig. 6 does not show which subfigure refers to which temperature). The manuscript should be carefully checked before submission.

2. The authors used different numbers of ethanol molecules to control the pressure. Please specify the exact initial pressure value for each simulation case (i.e. 50, 100, 150 and 200 ethanol molecules at 200 K).

3. The algorithm for calculation of adsorbed molecules could be

optimized. For the first adsorption mode, although the authors mentioned that the oxygen atom always heads to the surface Al atoms in the single-molecule adsorption case, other adsorptions such as Al-H could also happen for multi-molecule adsorption. For the second adsorption mode, the authors used the subtraction of total molecule number and number of ethanol molecules outside the adsorption region as the number of adsorbed molecules. However, this number includes the decomposed ethanol molecules, which should not be counted as the adsorbed ethanol molecules. Therefore, a better method to calculate the number of adsorbed ethanol molecules may be developed.

4. Page 11, Line 51-53: 'In the end, cases 65 ethanol molecules at most'. It is not clear why the maximum number of adsorbed ethanol molecules is 65. Based on Fig. 6b-6d (200 molecules case), the number of adsorbed ethanol molecules could continue to increase provided that the simulation time is long enough.

5. The authors mentioned in Section 3.3 (Page 6, Line 54) 'a timestep of 0.5 fs is enough', which indicates that the 0.5 fs timestep was used for all the MD simulations. But in Conclusion (Page 16, Line 36-37), the authors stated 'Simulation with timestep 0.1 fs'. Please clarify this.

6. The authors validated the ReaxFF force field used in this study by comparing its results of binding energies with DFT calculations. But is there any experimental evidence that can support the ReaxFF MD simulation results obtained from this research?

7. The adsorption is said to consist of three stages. In reality, the first two stages may not be distinguishable. The third stage is said to be zero. What does it mean?

8. Temperature is said to enhance adsorption (only) but temperature may also promote desorption too. Can the conclusion be of any general value, especially when the temperature range investigated is very limited.

Reviewer: 2

Comments to the Author(s)

This manuscript has a clear research topic and is suitable for Royal Society Open Science. Although the research is a little common and lack of innovation in some degree, the authors combine QM calculations and MD simulations in the process of ethanol's adsorption onto Al (111) under different temperatures and pressures. The employed methods and data analysis are both reasonable and scientific.

However, the pictures in this manuscript still have large space to be made better, such as Figure 1, the degree scales on vertical axis are too dense and small, same problems are also in Figure 5 and 8. The quality of pictures in Figure 5 and 8 is poor, the readers even cannot identify the details. Additionally, in Figure 10, there are no (a)-(d) in your RDF plots corresponding to figure caption. The other colored pictures, such as Figure 2 and 9 still need to be improved.

Overall, this work has scientific analysis and reasonable explanation for simulation phenomena, but some issues are still in this manuscript, a

major revision is necessary before reconsideration.

Some comments in details:

1. Your conclusion did not reflect your research result, in fact, your conclusion is only a copy of your abstract, especially the first half of Section Conclusion.

2. In the caption of Figure 8, I think that “different ethanol molecules at 300K” should be “different numbers of ethanol molecule”. Similar in caption of Figure 6, “Al slab at different temperature” should be “Al slab at different temperatures”, more important, what are different temperatures? Why not notice temperature values in figure caption? This kind of inappropriate description frequently appear in the manuscript, the authors should pay more attentions on English writing.

3. On page 13, line 51, the subheading “4.2.2 intermolecular struvture”, I think should be “intermolecular structure”, this kind of careless errors need to be avoided in the writing, especially in scientific manuscripts. The similar problems should be revised by the authors.

4. In Section 4.2.2, the authors mentioned “the most interesting $g(r)$ functions is the O-Al bonding”, in molecular dynamics simulations, the molecular and atomic interactions are common, RDF reflect the structure of atom (group) around one specific atom (group). However, how the authors can define the O-Al bonding in the adsorption of ethanol molecules on Al (111)? Can the authors explain it? Maybe under high temperature and pressure, there will be some chemical bonding forming. There is an obvious difference between physical interaction and chemical bonding.

5. In the description of Figure 6, when the number of ethanol molecule is 150, there is an obvious decreasing trend of adsorbed molecule number from 300K to 400K, while in cases of containing 200 ethanol molecules at 300K, 400K or 500K, the final adsorption number reach 65. How do the authors explain this abnormal phenomenon?

6. which indicates the Al slab used in this paper that can adsorb 65 ethanol molecules at most.

7. The cited references are old and less. Some important recent references need to be added. As far as we know, some related work about adsorption of single/multiple molecule(s) onto two-dimensional materials can be referred in your introduction or corresponding section, such as, (1) *Phys. Chem. Chem. Phys.*, 2017, 19(43): 29222-29231; (2) *Carbon*, 2014, 78: 500-509; (3) *J Phys Chem Lett*, 2018, 9(15): 4396-4400; (4) *The Journal of Physical Chemistry C*, 2018, 122(21): 11385-11391.

8. The format of cited references in your manuscript is inconsistent, some titles (Ref. 3-6) are capital at the beginning of a sentence, but some titles (Ref. 7, 10, 11, 14, 16) are capital letter in every word or for the whole of title.

Appendix B

Response to Reviewers

Dear editor, we have answered reviewer's comments item by item. The general improved content information is listed as follows:

To Reviewer 1:

Q1: Thanks for your careful inspection. We have rechecked the whole paper and some minor mistakes were corrected. All errors mentioned were also corrected.

Q2: In the new revised paper, the pressure information was listed in the second column of Table 3.

-----Initial Editor's/Reviewers' comments-----

Reviewer: 1

The revision is satisfactory and the paper is acceptable for publication after minor corrections. The detailed comments are:

Q1: The English and presentation of this manuscript have been improved. But there are still many errors. For example, Page 1, Line 48: keep them (their) chemical activity; Page 1, Line 50: Al_2O_3 , 2 & 3 should be subscripts; Page 2, Lines 15-16: there are still lack studies interpret interactions; Page, Line 17: 'detail' should be 'detailed'; Page 2, Line 27: We put our research focus on; Page 5, Line 34: are not ideal too (either); Page 5, Line 53: r_0 , p_{bo} , 0 & b_0 should be subscripts; Page 6, Line 16: E_{group} , E_{pair} , group & pair should be subscripts; Page 8, Line 21: 'begins' should be 'began'; Page 9, Line 30: there were two peaks appear; Page 18, Line 4: $R_{i+N/2}$, $i_{+N/2}$ should be subscript. I would suggest the authors do a thorough check and correct these errors.

Q2: Table 1 shows 'Atomic effective charges of ethanol model'. Where is the initial pressure information?

Q3: Revision accepted.

Q4: Revision accepted.

Q5: Revision accepted.

Q6: Revision accepted.

Q7: It was your question. I think the revision is fine.

Q8: It was your question. I think the revision is fine.

Reviewer: 2

No further comments.